# MECHANISTIC INTERPRETABILITY OF ANTIBODY LANGUAGE MODELS USING SAEs

**Rebonto Haque**
Department of Statistics
University of Oxford
Oxford, UK

**Oliver Turnbull**
Department of Statistics
University of Oxford
Oxford, UK

**Anisha Parsan**
Department of Electrical Engineering and Computer Science
Massachusetts Institute of Technology (MIT)
Cambridge, MA, USA

**Nithin Parsan**
Reticular
San Francisco, CA
USA

**John Yang**
Reticular
San Francisco, CA
USA

**Charlotte M. Deane**[*]
Department of Statistics
University of Oxford
Oxford, UK

## ABSTRACT

Sparse autoencoders (SAEs) are a mechanistic interpretability technique that have been used to provide insight into learned concepts within large protein language models. Here, we employ TopK and Ordered SAEs to investigate autoregressive antibody language models, and steer their generation. We show that TopK SAEs can reveal biologically meaningful latent features, but high feature–concept correlation does not guarantee causal control over generation. In contrast, Ordered SAEs impose a hierarchical structure that reliably identifies steerable features, but at the expense of more complex and less interpretable activation patterns. These findings advance the mechanistic interpretability of domain-specific protein language models and suggest that, while TopK SAEs suffice for mapping latent features to concepts, Ordered SAEs are preferable when precise generative steering is required.

## 1 INTRODUCTION

Antibodies are a key part of the body's adaptive immune response and are characterised by their ability to bind to a specific antigen and subsequently neutralise it or initiate an immune response. They possess significant sequence, and therefore structural, diversity, which enables binding to virtually any target antigen (Chiu et al., 2019).

The antigen-binding domain of antibodies consists of variable heavy (VH) and variable light (VL) chains, whose binding specificity and affinity are primarily determined by six complementarity-determining regions (CDRs), three on each chain. Numerous V (variable), D (diversity), and J (joining) gene segments in the genome encode these chains, with combinatorial assembly of V, D, and J for VH and V and J for VL generating substantial sequence diversity. This diversity is further enhanced by somatic hypermutation, in which random nucleotide substitutions occur at markedly elevated rates within the rearranged V(D)J segment (Andreano & Rappuoli, 2021).

The ability to bind any target antigen with high specificity and affinity makes antibodies ideal candidates for drug discovery. As a result, antibody drugs hold a major and growing share of the total pharmaceutical market (Crescioli et al., 2025). Antibody drug development pipelines need to identify candidates which bind specifically and with high affinity to the target antigen, while also being 'developable' (Jarasch et al., 2015). 'Developability' refers to properties required for

---

[*]Correspondence: deane@stats.ox.ac.uk

a successful drug such as immunogenicity, solubility, specificity, stability, manufacturability, and storability (Raybould & Deane, 2022).

Antibody language models have been used to optimise multiple steps of antibody-drug development pipelines from library generation (Turnbull et al., 2024a) to humanisation during lead optimisation (Chinery et al., 2024). p-IgGen is a GPT-like decoder-only model trained on paired antibody-sequence data, consisting of 17M parameters (Turnbull et al., 2024a). ProGen2-OAS (Nijkamp et al., 2023) is a larger, architecturally similar, decoder-only model ( 764M parameters) trained on unpaired antibody-sequence data. IgLM (Shuai et al., 2023) similarly uses a GPT-like decoder-only architecture with 13M parameters, but was trained adopting the infilling language model formulation (Donahue et al., 2020) using unpaired antibody-sequence data.

The lack of interpretability of machine learning models contributes to a lack of trust in model predictions, difficulty determining whether biologically relevant features are being used to make predictions and difficulty detecting overfitting. Collectively, these pose a barrier when employing language models for drug discovery (Chen et al., 2023). SAEs offer a promising approach to identify human-interpretable concepts learned by models and steer their generation (Chen et al., 2025; Templeton et al., 2024). Prior works have used SAEs to understand the inner mechanisms of PLMs (Adams et al., 2025; Parsan et al., 2025; Simon & Zou, 2024), and steer model output. However, to date, SAEs have not been used to interrogate autoregressive protein or antibody-specific language models.

This work advances the interpretability of antibody language models, using SAEs to identify biologically relevant features of interest learned by the antibody language models p-IgGen, IgLM, and ProGen2-OAS and predictably steer their generation. We identify antibody-specific features, such as the complementarity-determining region (CDR) identity and germline gene identity, and use them to steer model generation for specific germline gene identities. Overall, this work shows the applicability of SAEs for incorporating rational design principles to antibody library generation, allowing the generation of libraries with desired properties such as enrichment in sequences originating from specific germlines. We show that TopK SAEs can accurately identify interpretable latents underpinning model generation, whereas Ordered SAEs can identify steerable features capable of tuning model generation.

## 2 RELATED WORK

### 2.1 MECHANISTIC INTERPRETABILITY

Mechanistic interpretability refers to the approach of explaining complex machine learning systems through the behaviour of their functional units (Kästner & Crook, 2024) by decomposing or reverse-engineering systems into their more elementary computations (Rai et al., 2025). The eventual goal is to discover causal relationships between model inputs and corresponding outputs.

### 2.2 SPARSE AUTOENCODERS

Sparse Autoencoders (SAEs) have specifically been employed in mechanistic interpretability for feature discovery. They tackle the issue of feature superposition resulting in polysemantic neurons, where any given neuron encodes multiple, often unrelated features. SAEs tackle this problem by projecting dense neuron activations into a sparser latent space using a sparse encoder, Equation 1, whilst ensuring the latent representation can be reconstructed back into the original neuron representation by a decoder following sparsification, Equation 2.

$$z = g\big(ReLU\big(W_{\text{enc}}\,x + b_{\text{enc}}\big)\big) \tag{1}$$

$$\hat{x} = W_{\text{dec}}\,z + b_{\text{dec}} \tag{2}$$

where $W$ are the weight matrices and $b$ are the bias vectors, enc and dec denote the encoder and decoder respectively, $x$ is the original hidden representation, $z$ the latent representation, and $\hat{x}$ the reconstructed hidden representation. $ReLU$ activation is applied to the latent representation following encoding and $g$ is a sparsification function.

### 2.2.1 TOPK SAEs

TopK SAEs (Gao et al., 2024) limit the number of active latents to $k$, where $k \ll d_{\text{in}} \ll d_{\text{sae}}$. $d_{\text{in}}$ is the input hidden dimensions, and $d_{\text{sae}}$ is the latent or dictionary dimensions. Equation 3 shows the loss computation.

$$L(x) = \underbrace{\|x - \hat{x}\|_2^2}_{\text{Reconstruction loss}} + \underbrace{c}_{\text{Sparsity constraint}} \tag{3}$$

The $L(x)$ reconstruction loss compares the decoded representation $\hat{x}$ with the original hidden representation $x$. When a sparsification function is not directly applied during encoding, a separate sparsity constraint is added in loss computations, which is usually a variation of an L1 regularisation loss (Zhang et al., 2018).

### 2.2.2 ORDERED SAEs (O-SAEs)

Ordered SAEs follow a nested SAE architecture, enabling hierarchical ordering of SAE latents. Importantly, compared to the traditional TopK SAE architecture which arbitrarily orders hierarchical latents within the dictionary space, O-SAEs enforce a strict, consistent, hierarchical ordering of latents. This is because TopK SAEs enforce sparsity within the entire dictionary space in one go, whereas O-SAEs follow a nested approach and effectively train a number of individual, nested SAEs which occupy an increasing portion of the dictionary space.

O-SAEs introduce two core components: (i) *per-index nested grouping*, and (ii) *strictly decreasing truncation weights* in order to ensure consistent ordering.

(i) For each truncation level $m \in \{1, \ldots, d_{\text{sae}}\}$, the first $m$ rows of the encoder and decoder are isolated:

$$W_{\text{enc}}^{(m)} = [W_{\text{enc}}]_{1:m,\,:}, \quad W_{\text{dec}}^{(m)} = [W_{\text{dec}}]_{1:m,\,:} \tag{4}$$

In Eq. (4) the encoder–decoder pair $\left(W_{\text{enc}}^{(m)}, W_{\text{dec}}^{(m)}\right)$ re-uses the first $m$ rows of the full weight matrices. Because every smaller autoencoder is a strict subset of the larger one, any latent $i \leq m$ is shared across all groups that follow. This "per-index nested grouping" forces early latents to model global structure that remains useful for every deeper stage. Per-index grouping ensures non-random sampling of dictionary sizes, unlike in Matryoshka SAEs (Bussmann et al., 2025), increasing the overall consistency of results.

(ii) Each partial reconstruction is weighted by a *monotonically decreasing* probability $p_M(m)$, so that early (low-index) features incur a higher penalty when failing to capture coarse structure. The per-truncation loss is

$$L_m(x) = p_M(m) \left\| x - W_{\text{dec}}^{(m)\top} W_{\text{enc}}^{(m)} x \right\|_2^2 \tag{5}$$

and summing over all $m$ promotes the model to learn the most "abstract" elements first, with progressively finer details later. Combining the decreasing probability weights with nested latents further enforces ordering of identified latents and maintains a stricter hierarchy.

## 3 DATA AND METHODS

### 3.1 SPARSE AUTOENCODER TRAINING

We adapted TopK Sparse Autoencoders (SAEs) from the EleutherAI/sparsify GitHub repository (https://github.com/EleutherAI/sparsify), and Ordered SAEs based on (Wang et al., 2025). Training parameters were taken directly from the original repositories where available.

### 3.1.1 BASE MODELS AND ACTIVATION EXTRACTION

Our primary experiments use p-IgGen, an autoregressive antibody language model trained on paired OAS. To test robustness, we also run the same SAE training and concept-probing pipeline on IgLM and ProGen2-OAS, which differ in architecture and scale. We trained both the TopK and Ordered SAEs on hidden layer activations of respective models, generated from the original p-IgGen training set (Turnbull et al., 2024b). For p-IgGen, we extract activations from each of its 4 hidden layers. For IgLM and ProGen2-OAS we only extract activations from their final layers for ease of comparison

across models. This is mainly because ProGen2-OAS has 27 layers, whilst p-IgGen and IGLM have 4.

The training set contained **1,800,545** VH/VL paired sequences from the Observed Antibody Space database (OAS) (Olsen et al., 2022; Kovaltsuk et al., 2018). When generating p-IgGen activations for training, we concatenated the paired VH and VL sequences together, with appropriate start and end tokens added, and passed them into p-IgGen to generate hidden activations. This generated 4 sets of hidden activations, one from each hidden layer. For Ordered SAE training, we randomly subsampled **100,000** sequences to decrease training time.

We further trained SAEs on final hidden layer activations of IgLM and ProGen2-OAS using the same p-IgGen training sets as before. Since IgLM and ProGen2-OAS were originally trained using unpaired sequences, we generated activations for unpaired VH and VL chains with appropriate start, end, and padding tokens from the respective vocabularies.

### 3.1.2 TOPK SAE

The model's input dimensions $d_{\text{in}}$ were projected onto a higher-dimensional latent/dictionary size $d_{\text{sae}}$, where $d_{\text{sae}} = d_{\text{in}} \times r = d_{\text{in}} \times 32$. $r = 32$ is the expansion factor. ReLU activation was applied to the projection, $z = \text{ReLU}(W_{\text{enc}} x + b_{\text{enc}})$, followed by a Top-$k$ sparsification with $k = 32$, retaining only the top 32 activations by magnitude. The resulting dictionary size was **32x**. Decoder weights were initialised as the unit-normalised transpose of the encoder weights to stabilise training. Training used a batch size of 8 and Adam optimisers throughout, with a custom learning rate $\eta = \frac{2 \times 10^{-4}}{\sqrt{d_{\text{sae}}/16{,}384}}$.

### 3.1.3 ORDERED SAE

Ordered Sparse Autoencoders (O-SAEs) were adopted to retain higher-level, abstract features within our latent space and hierarchically arrange the latents. In our setup, we used expansion factor $r = 8$, yielding a dictionary size $d_{\text{sae}} = d_{\text{in}} \times r = d_{\text{in}} \times 8$. Sparsity was again set to $k = 32$, ensuring the top 32 latents are used during reconstruction. All models were trained with Adam optimisers at a fixed learning rate $\eta = 1 \times 10^{-4}$. We chose a smaller maximum dictionary size for the O-SAEs to speed up training, effectively reducing the total number of nested SAEs being trained. Due to per-index grouping, O-SAEs need to train several nested SAEs based on the total dictionary size, whereas the regular TopK architecture only trains a single model.

### 3.2 TARGETED FEATURE IDENTIFICATION USING SAEs

### 3.2.1 TRAINING DATA

Paired antibody sequence data were obtained from OAS, Coronavirus Antibody Database (CoV-AbDab) (Raybould et al., 2021), and the Patent and Literature Antibody Database (PLAbDab) (Abanades et al., 2024). A total of 149,069 sequences were obtained from the respective datasets, based on their binding specificities to SARS-CoV2 RBD (binder and non-binder).

The data was clustered based on CDR sequence similarity using CD-HIT (Li et al., 2001), with a 0.8 similarity threshold on the total CDR sequence. The clusters were then randomly split into the training-validation-test set, whilst ensuring members of the same cluster were in only one of the three possible splits. The splits were further stratified based on binding specificity to SARS-CoV2 RBD.

This specific dataset was originally prepared for a separate project, and the SARS CoV2 RBD binding properties of the antibodies are not relevant for this study. Qualitatively, a dataset of equivalent size randomly sampled from OAS should produce the same results.

The following concepts were studied to identify associated latents: CDR identity, which refers to whether a given residue lies within a specific CDR region, and V/J gene identity, which refers to the germline V or J gene segment that was used to code for the final antibody sequence. For the CDR-identity, the training matrix was the latent activations for each residue. The CDR identity dataset had 7 classes (6 CDR identities and non-CDR regions). For sequence-level concepts (V/J gene identity), the sequence representation used for the linear probe was computed by mean-pooling the *non-zero* residue-level latent activations in each sequence (i.e., averaging only over positions

where the latent is active). For the activation-threshold analysis, we instead used mean-pooling over *all* residue positions (including zeros) to obtain a consistent sequence-level summary for thresholding, similar to previous work (Adams et al., 2025; Simon & Zou, 2024).

### 3.2.2 LINEAR PROBE

We trained a logistic regressor to act as a linear probe on the training-validation data. A logistic regressor (LR) was trained, employing 3-fold cross-validation grid search to optimise hyperparameter C. In logistic regression, C is the inverse of the regularisation strength: larger C applies less regularisation and can overfit, while smaller C applies more regularisation and can improve generalisation. Cross-validation was done during training by randomly shuffling and splitting the training data into 3 cross-validation sets. Correlation weights of all latents were stored and the latents with the top 500 positive correlation weights were used for further validation.

### 3.2.3 LATENT SELECTION

The top correlated latents were further validated on the validation set. Based on the strategy by Simon and Zhou (Simon & Zou, 2024), each latent was normalised across the validation set using MinMax scaling. In this instance, similar to Simon and Zhou, we took the mean activation of the latents across *all* residue positions in the sequence. This all-position mean-pooling is used for the activation-threshold analysis; the linear probe instead uses non-zero mean-pooling as described above. For each normalised latent, we constructed binary latent-on/latent-off labels using activation thresholds of 0.1, 0.2, 0.5, 0.8, and 0.9. For each latent–concept pair, we defined a latent as an interpretable feature only if it achieved an $F_1$ score that exceeded a class-prevalence baseline by a fixed margin. For each concept class (treated one-vs-rest), let $\pi$ denote the prevalence of that class in the validation set. We first computed the $F_1$ score of a trivial always-positive classifier,

$$F_{1,\text{all+}} \;=\; \frac{2\pi}{1+\pi}. \tag{6}$$

A latent was labelled a feature if, for any tested activation threshold, its $F_1$ satisfied

$$F_1 \;>\; F_{1,\text{all+}} + 0.2 \tag{7}$$

This criterion accounts for class imbalance and requires each latent to provide predictive signal beyond a strong non-informative baseline, rather than exploiting the marginal class distribution. We use 0.2 as a conservative margin to reduce false discoveries. It is worth noting that the threshold for 'feature' is somewhat arbitrary. Latents just below the threshold are likely to still have meaningful correlations to a target concept.

### 3.2.4 ANTIBODY SEQUENCE ALIGNMENT

Antibody sequences were aligned using ANARCI (Dunbar & Deane, 2016) and the IMGT numbering (Lefranc et al., 2003).

### 3.3 STEERING

Steering was implemented based on the strategy by Templeton et al. (Templeton et al., 2024). Each latent can be represented by its corresponding decoder vector $d(i) = W_{\text{dec}}[i, :]$, where $d(i)$ is the decoder vector for latent $i$ and $W_{\text{dec}}$ is the decoder weight matrix. Steering is performed by scaling the decoder vector and adding it to the original hidden state (Equation 8).

$$h_l^* \;\leftarrow\; h_l \;+\; \alpha \cdot d(i) \tag{8}$$

Here, $\alpha$ is the steering factor and $h_l$ is the hidden state before the intervention and $h_l^*$ is the hidden state following the intervention.

### 3.4 ANALYSING NOVELTY, DIVERSITY, AND ANTIBODY-LIKENESS OF GENERATED SEQUENCES

To assess whether nested steering alters the generative distribution while retaining antibody-like properties, we evaluated steered libraries using sequence novelty (Hamming distance to the held-out validation set), diversity (nearest-neighbour cosine distance), and protein-likeness using ESM-2 likelihood scores (Lin et al., 2023).

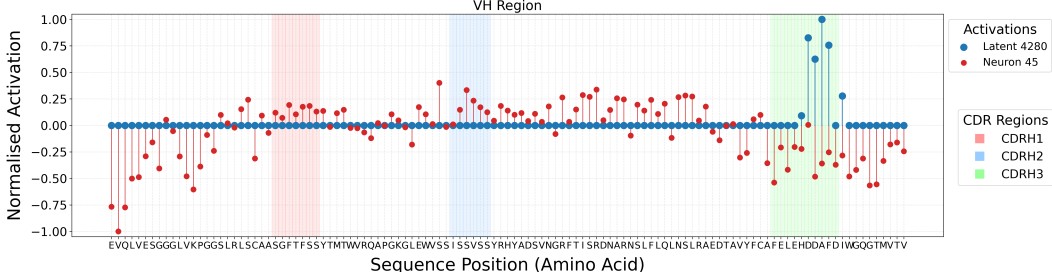

Figure 1: Top-K latent activations (blue) and hidden neuron activations (red) for CDRH3 identity collected from p-IgGen. The x-axis shows the amino-acid sequence of the VH region of a test antibody; the y-axis shows normalised activation. CDRs are coloured CDRH1 (red), CDRH2 (blue), and CDRH3 (green). Latent activations localise to the CDRH3 loop, whereas neuron activations are scattered across the sequence with no discernible pattern.

## 4 RESULTS

### 4.1 TOPK SAE LATENTS PRESERVE BIOLOGICAL INFORMATION FOLLOWING SPARSIFICATION

TopK sparsification represents each token with far fewer latents than hidden neurons, raising the possibility of information loss, so we compared latents and hidden neurons on residue- and sequence-level property prediction tasks. For TopK SAEs trained on final-layer activations (layer 3) (Parsan et al., 2025), logistic regressors trained on CDR identities achieved validation accuracies of **0.99** using latent activations and **0.98** using hidden neuron activations, indicating that residue-level information is preserved. For sequence-level features, germline heavy J gene prediction using latent activations yielded a validation F1 macro score of **0.93** (reported due to class imbalance), with Supplementary Table S3 showing strong precision, recall, and F1 across IGHJ classes. Together with similar findings for general protein language models (Simon & Zou, 2024; Adams et al., 2025; Parsan et al., 2025), these results suggest that SAE latents preserve key antibody information after sparsification, justifying their use for further interpretability analysis.

### 4.2 TOPK LATENT ACTIVATIONS ARE VISUALLY INTERPRETABLE

To investigate whether SAE latents provide an interpretable alternative to understanding model generation, we compared the activated patterns of latents and neurons correlated to properties of interest. As a baseline, we compared activations of the top correlated latents and neurons for CDRH3. Visual investigation reveals latent activations are sparse and specific to CDRH3 residues, compared to neurons which activate across the sequence without any immediately recognisable pattern (Figure 1). This may be explained by the polysemanticity of neurons, where multiple features specific to several unrelated residues are represented by the same neuron. When investigating activation patterns, this complicates using neurons as a tool for interpretability and highlights the potential greater explainability of SAE-derived latents.

We further investigated heavy J gene activations as sequence-level concepts. Latents corresponding to heavy J gene identity activated on residues representing the concept, i.e. gene identity. In this instance, the top correlated latents were activated on the J domain of examined antibodies (Figure 2). This is interesting because, although the sequence-level probe uses mean-pooled activations, the underlying residue-level latents still localise to the J region when examined positionally. To quantify the predictive properties of our identified features, we carried out an activation-threshold analysis (Supplementary Table S2).

### 4.3 ORDERED SAEs IDENTIFY MORE STEERABLE FEATURES COMPARED TO TOPK SAEs

In addition to assessing how well latents predict the target concept, we also use feature steering as an indicator of feature importance (Parsan et al., 2025): if increasing a latent consistently steers generation in a desired direction, the corresponding feature is likely important. We were unable to successfully steer on TopK latents within p-IgGen's latent space (See Appendix A), which may be

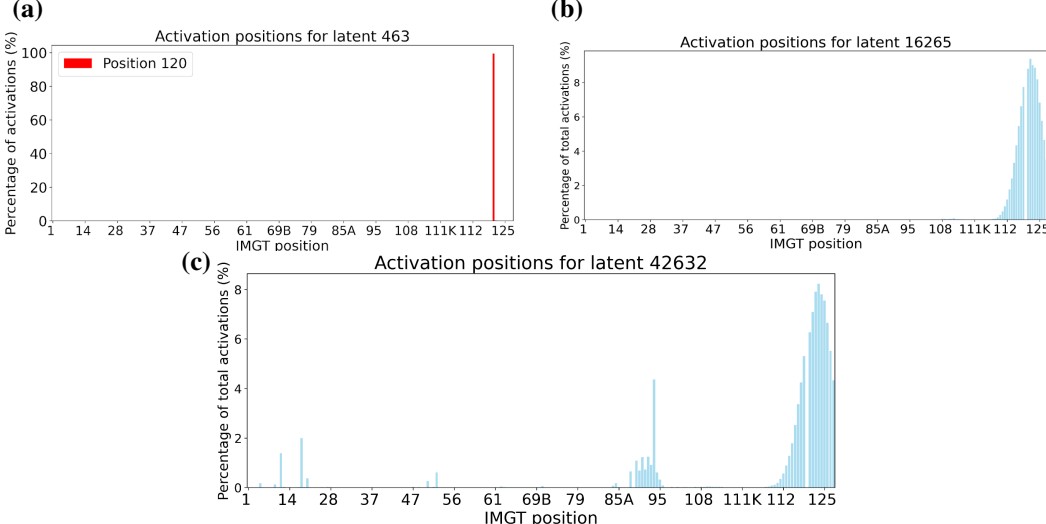

Figure 2: IMGT-aligned Top-K latent activation profiles in the validation set for the top IGHJ4-associated latent from an SAE trained on (a) p-IgGen activations, (b) IgLM activations, and (c) ProGen2-OAS activations. The x-axis shows IMGT positions and the y-axis shows the fraction (or normalised magnitude) of activations at each position aggregated across validation sequences.

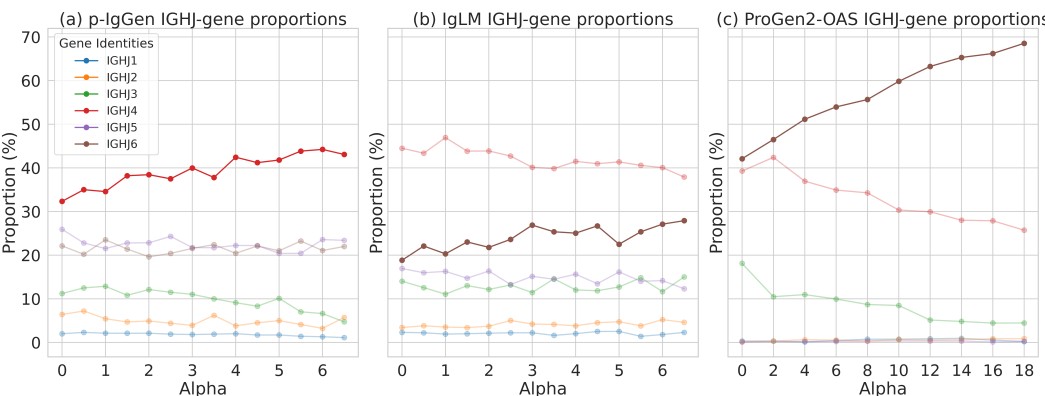

Figure 3: Results of IGHJ feature steering using O-SAE latents for (a) p-IgGen latent 12, positively correlated with IGHJ4, (b) IgLM latent 85, positively correlated with IGHJ6, and (c) ProGen2-OAS latent 119, positively correlated with IGHJ6. The y-axis shows the proportion of generated sequences. Plots are coloured by heavy J gene identity, with the targeted gene identity highlighted. The x-axis shows the steering factor used ($\alpha$). Results are for a library of 1000 generated sequences.

attributed to known issues within these architectures such as feature splitting and absorption (Chanin et al., 2024). We subsequently evaluated Ordered SAEs, which build a hierarchical latent space that preserves both high-level and fine-grained features (Bussmann et al., 2025), but result in a less interpretable localisation pattern (Supplementary Figure S3).

We conducted a linear probe and subsequent activation-threshold analysis to identify features correlated to IGHJ4 in layer 3. Due to the implicit hierarchy in O-SAE features, we first filtered to latents with high $F_1$ scores and positive probe weights, then prioritised lower dictionary indices (which correspond to higher-level features). We identified latent 12 within p-IgGen's latent space which was positively correlated to IGHJ4 and used it to steer model generation (Figure 3 a). To test the robustness of the steering approach we also steered IgLM using latent 85 (Figure 3 b) and ProGen2-OAS using latent 119 (Figure 3 c), both of which were positively correlated with IGHJ6.

Positively steering on latent 12 increased IGHJ4 proportion in p-IgGen generation (Spearman $\rho = 0.921$, $p = 2.982 \times 10^{-6}$). For ProGen2-OAS steering, correlations between the target IGHJ6 score

and the steered latent activation were highly significant (Spearman $\rho = 1.000$, $p = 6.647 \times 10^{-64}$). For IgLM steering, we likewise observed significant correlations, though with smaller magnitude (Spearman $\rho = 0.802$, $p = 5.563 \times 10^{-4}$). Subsequent analysis indicated that steering results in sequences that are novel, diverse, and antibody-like (Supplementary Table S4).

## 5 Conclusions and Future Outlook

Sparse autoencoders offer a practical route to interrogate autoregressive antibody LMs: they surface domain-specific features, but high correlation does not guarantee causal control. In our study, TopK SAEs often produced interpretable, residue-level signals that were not steerable, whereas Ordered SAEs yielded more abstract—and reliably steerable—features at the cost of intuitive localisation. Progress is currently limited by scarce labelled antibody datasets, which hampers systematic feature discovery and validation. For studies on antibody language models, we advocate rigorous steering/ablation benchmarks and scaling SAE training with curated, annotated resources (e.g., FLAb) to learn higher-level, controllable features. Realising this will enable targeted manipulation of properties such as developability and specificity, advancing rational, model-guided antibody library design.

MEANINGFULNESS STATEMENT

Life is organised through molecular sequences, and protein and antibody language models aim to learn this organisation from data. Yet their opacity makes it unclear whether they capture real biological structure or spurious correlations. We use sparse autoencoders to reveal antibody-relevant concepts represented inside autoregressive antibody language models and to test whether these concepts causally affect generation. We find that some features are interpretable but not reliably controllable, while ordered representations identify features that support predictable steering. By making internal representations experimentally actionable, this work helps open the black box and extract more reliable biological insight from language models.

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

APPENDIX

## A  CASE STUDY OF HEAVY J GENE IDENTITY FOR IGHJ4

The functional importance of SAEs for studying LLMs lies in their ability to interrogate specific, domain-relevant concepts. Here, we examine gene identity, as it directly influences antibody binding affinity and specificity (Deng et al., 2025). Specifically, we pick IGHJ4 genes for our analysis, due to their widespread clinical significance underpinned by the fact that they are the most widely utilised J genes in our immune repertoire. That is, the majority of heavy chain antibodies within any given individual originate from the IGHJ4 germline gene. For our analysis, we decided to focus on the final

Table S1: IGHJ4 feature statistics across p-IgGen layers. A latent is counted as a feature if it satisfies $F_1 > F_{1,\text{all}+} + 0.2$ for any tested activation threshold.

| IGHJ | Layer | Features | Max $F_1$-score |
|---|---|---|---|
| IGHJ4 | Layer 0 | 1 | 0.930 |
| | Layer 1 | 4 | 0.949 |
| | Layer 2 | 4 | 0.930 |
| | Layer 3 | 1 | 0.752 |

layer (Table S1).

First, we looked at the absolute positional activations of this latent across all the sequences in our validation set which had an IGHJ4 heavy J gene, and compared it to activations on specific IMGT positions. Whilst activations on absolute positions were distributed near the end of the heavy chain corresponding to the J region, the activations on IMGT positions were more consistent and concentrated. This implies that the model does not base its activation pattern on the absolute sequence length alone, but rather the underlying sequence alignment (Figure S1).

We then chose to investigate the specific residue identities on which the latents were activated. Based on the heavy J gene sequence alignments, the top two latents activated at IMGT positions 120 and 119, which are a Q and G, respectively. These are conserved across all human IGHJ genes. The third top latent activated on Y at position 117, which is unique for IGHJ4 (Scaviner et al., 1999). These results indicated that top latents encoded contextual information of the preceding residues.

Previous studies have highlighted how highly correlated features may be used to steer model outputs (Templeton et al., 2024; Simon & Zou, 2024). We attempted to steer on each identified feature to investigate how it affects model generation. We positively steered on each latent, which we hypothesised should increase the proportion of IGHJ4 in generated sequences. However, steering on these latents was unpredictable and did not consistently increase IGHJ4 proportions (Figure S2).

To check if this phenomenon was somehow exclusive for IGHJ4 and layer 3, we attempted to steer across all the layers for a number of different features for various gene identities, but were unable to predictably steer model generation (data not shown). The lack of steerability may indicate how these features individually do not contribute to the gene identity, making them informative features when used for downstream predictions, but not for biasing model output.

This may be due to feature splitting (Chanin et al., 2024) which has been reported for TopK SAEs. Feature splitting refers to the phenomenon where higher-order features are broken down into specific contextual examples. In the case of text-based language models, 'math' may be split into 'algebra' and 'geometry'. These phenomena arise when enforcing sparsity in a dictionary consisting of hierarchical features. In this instance, if the identified latents correspond to only single residues within the J-domain, it essentially becomes a residue-level feature as opposed to a sequence-level feature. If the feature activates on a residue specific to the gene identity, it may be a good predictor for the gene identity, but not a steerable feature. This points to the possibility that several features together confer J gene identity, and that these features are likely correlated to each other. Hence, activating one but not the others does not necessarily result in a predictable shift in model output.

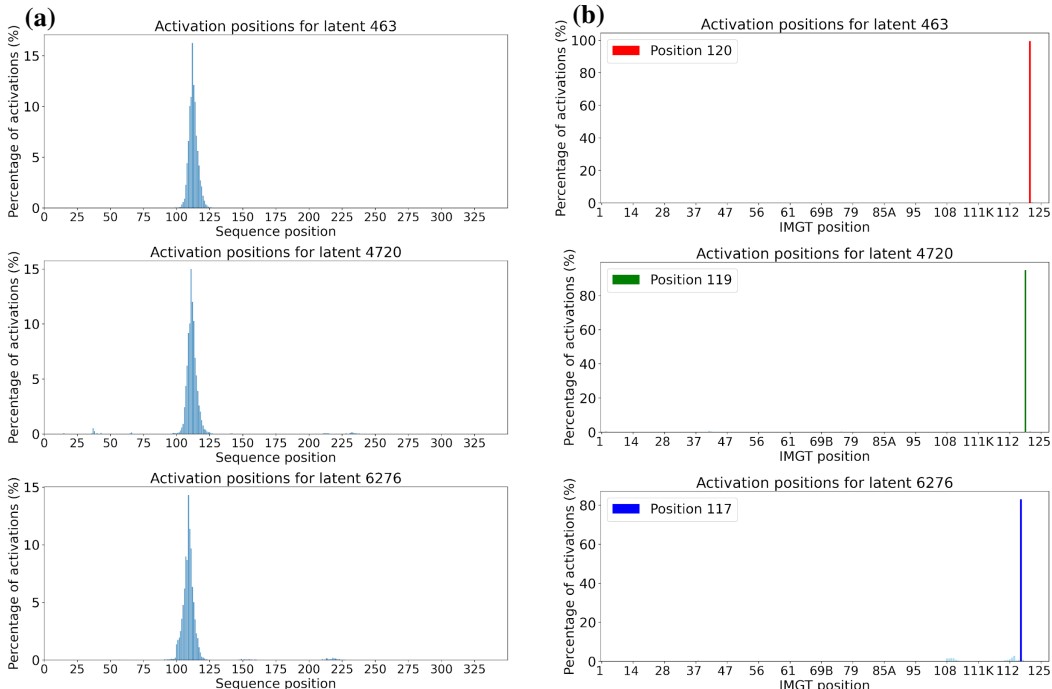

Figure S1: Comparison of absolute positional (a) and IMGT (b) activations of top three IGHJ4 Top-K latents. The sequence/IMGT positions are shown on the x-axis. For the sequence positions, the amino acid sequences were end-padded to a constant length of 350. Percentage of total activations on any given position across validation IGHJ4 sequences is shown on the y-axis. The most frequent IMGT position for activation is highlighted for each latent. Latent activations show a distribution near the end of the heavy chain when aligned based on absolute sequence position. In contrast, latents demonstrate discrete activations when aligned based on IMGT numbering.

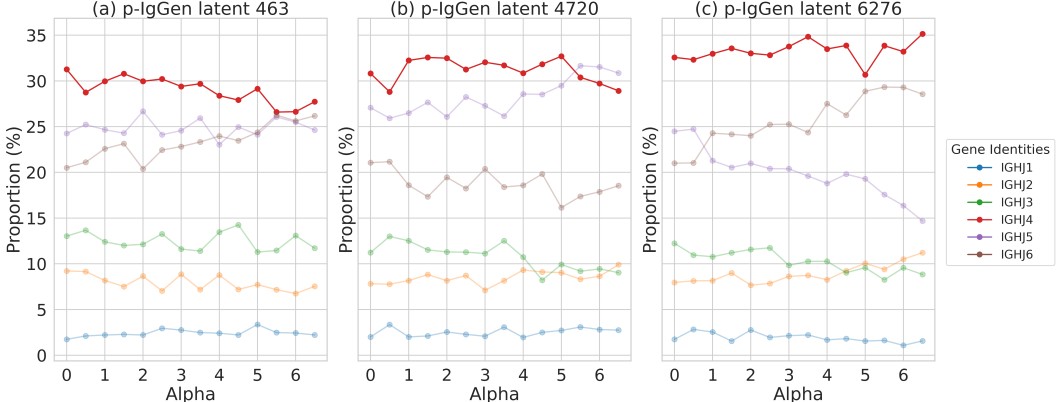

Figure S2: Results of IGHJ4 feature steering for p-IgGen Top-K latent 463 (a), 4720 (b), 6276 (c). Y-axis shows the proportion of generated sequences. Plots are coloured by heavy J gene identity. X-axis shows the steering factor used (alpha). Results are for a library of 1000 p-IgGen-generated sequences. For each latent tested (a-c), steering did not result in a predictable change in library composition.

The case study on IGHJ4 indicated that identified features retain biologically relevant context information. Most highly correlated features (based on LR correlation weight and $F_1$ score) tend to be residue-specific. Targeted approaches such as this cannot easily find abstract, higher-order features, assuming they are represented within the latent space to begin with. Concept-specific targeted feature identification might identify highly correlated features that are biologically informative. For instance,

two of the three top features (463 and 4720) activate on conserved residues preceded by sequence motifs specific to the gene identity. The third, Latent 6276 activated on an IGHJ4-specific residue, which may explain why this feature can be used to accurately identify IGHJ4.

Highly predictive features may be correlated with other biologically informative features. To understand whether highly predictive features influence model behaviour, we tried to steer along these features to increase the proportion of IGHJ4 in generated sequences. This did not produce predictable results, making it difficult to interpret the contribution of each individual latent to model generation. Overall, TopK SAEs can identify features in targeted concept analysis which are intuitively interpretable, however, not necessarily steerable.

## B  SUPPLEMENTARY FIGURES AND TABLES

Table S2: Layer 3 feature counts and maximum $F_1$-scores for IGHJ genes. A latent is counted as a feature if it satisfies $F_1 > F_{1,\text{all}+} + 0.2$ for any tested activation threshold, where $F_{1,\text{all}+} = \frac{2\pi}{1+\pi}$ and $\pi$ is the validation prevalence of the predicted class in a one-vs-rest setting.

| Gene | Number of features | Max $F_1$ | $F_{1,\text{all}+} + 0.2$ |
|---|---|---|---|
| IGHJ1 | 1 | 0.366 | 0.244 |
| IGHJ2 | 6 | 0.758 | 0.393 |
| IGHJ3 | 17 | 0.866 | 0.364 |
| IGHJ4 | 1 | 0.752 | 0.713 |
| IGHJ5 | 0 | 0.486 | 0.598 |
| IGHJ6 | 16 | 0.866 | 0.517 |

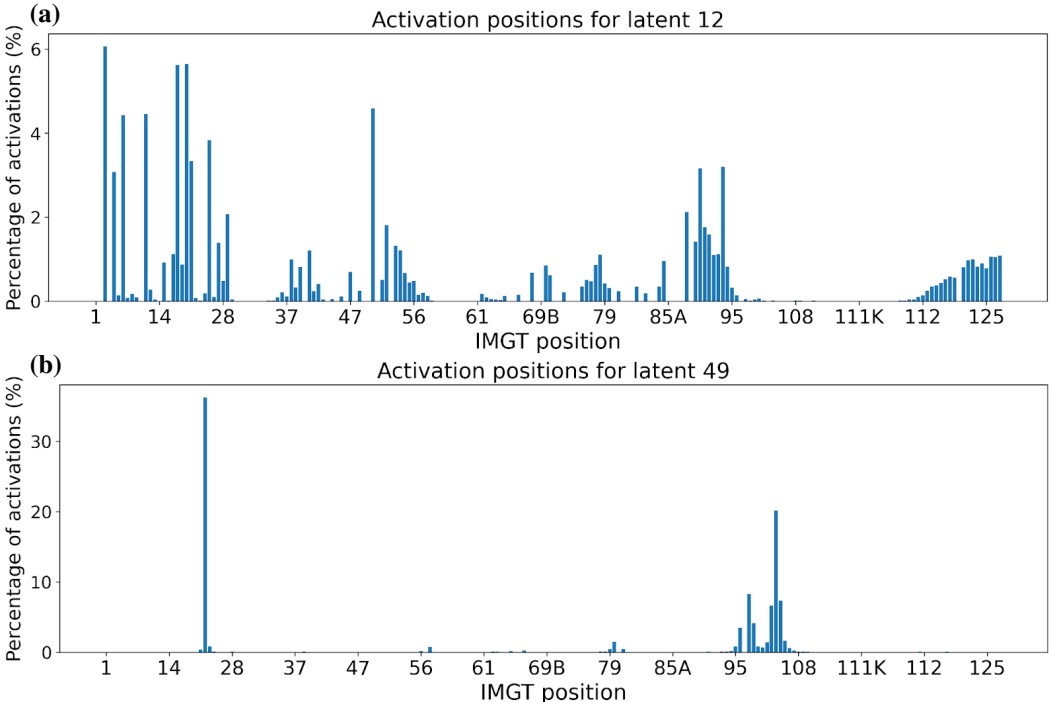

Figure S3: IMGT activations of p-IgGen O-SAE latent 12 (a) and 49 (b). Activation patterns of both latents show scattered distribution across the range of IMGT positions.

Table S3: Precision, recall, and $F_1$-score per IGHJ class.

| Gene | Precision | Recall | $F_1$-score |
|------|-----------|--------|-------------|
| IGHJ1 | 0.91 | 0.71 | 0.80 |
| IGHJ2 | 0.94 | 0.93 | 0.94 |
| IGHJ3 | 0.96 | 0.96 | 0.96 |
| IGHJ4 | 0.98 | 0.99 | 0.98 |
| IGHJ5 | 0.94 | 0.95 | 0.95 |
| IGHJ6 | 0.98 | 0.97 | 0.98 |
| Macro average | 0.95 | 0.92 | 0.93 |

Table S4: Asterisks denote statistically significant differences between steered and natural sequences (two-sided Mann–Whitney $U$ test, $p < 0.05$). Steering with O-SAE latent 12 at $\alpha = 6.50$ slightly increased novelty (VH: 24.67 vs. 22.91; VL: 20.24 vs. 19.81) and diversity (0.23 vs. 0.20), while protein-likeness was essentially unchanged (ESM-2 likelihood $-0.325$ vs. $-0.330$).

| Metric | Steered | Natural | $\Delta$ |
|--------|---------|---------|----------|
| Mean VH Hamming dist. to val. set | 24.67* | 22.91 | 1.76 |
| Mean VL Hamming dist. to val. set | 20.24* | 19.81 | 0.43 |
| Diversity | 0.23* | 0.20 | 0.03 |
| ESM2-likelihoods | $-0.325$* | $-0.330$ | 0.005 |

