# OpenReview forum: "Mechanistic Interpretability of Antibody Language Models Using SAEs"
_ICLR.cc/2026/Workshop/LMRL — ICLR 2026 Workshop LMRL Poster_

### Official Review · Reviewer_ifeS · 2026-02-11

**Rating:** 8
**Confidence:** 2

**Review:**

## Summary
This manuscript describes a method that utilizes Sparse Autoencoders to interpret Antibody Language Models including p-IgGen, IgLM, and ProGen2-OAS. The research focuses on TopK and Ordered SAE, which identified more visualizable and steerable features independently.

## Strengths
1. This paper explains the black box nature of deep learning, which illuminates how deep learning model understanding antibody modeling problems. It enables us to explore potential underly mechanism and build truthworthable models.
2. It discussed two types of SAE, which demonstrates how different methods interpret the same problem.

## Weaknesses
1. The authors use 0.2 as an arbitrary threshold, which is acceptable for a automated analysis. But discussing this threshold is still helpful.
2. As an interpretability work, it will be more solid to add quantitative perturbation evaluation such as LOdds and AOPC to check whether the interpretation is faithful or not.
3. If there could be some case studies to use ribbon diagrams and align the activation scores to the figure, the visualization may be more informative.

---

### Official Review · Reviewer_LuGC · 2026-02-16
**Review on SAEs for antibody language models**

**Rating:** 7
**Confidence:** 5

**Review:**

This paper applies state-of-the-art mechanistic interpretability methods (sparse autoencoders, SAEs) to antibody language models. It compares two different types of SAE architectures (Ordered SAEs and TopK SAEs), and investigates whether these can help identify interpretable and steerable latents.

Overall, the paper is well structured and well written. The research question is well motivated, and the authors explore a variety of models to answer it. There are some interesting results, with Ordered SAEs being better at extracting steerable features and TopK SAEs producing more interpretable features. The methodology is clear, and the authors account for class imbalance and sequence clustering based on protein similarity to reduce data leakage. They also quantify steering effects using Spearman’s correlation.

However, there are a few weak points and questions:

- The authors analyze all layers of p-IgGen and only the final layers of IgLM and ProGen2-OAS. It is not clear what motivated this layer choice, especially since it is common practice in other papers to use middle layers.
- The authors do not mention some relevant papers from the literature on SAEs in protein language models (e.g., https://arxiv.org/abs/2502.09135 and https://arxiv.org/abs/2509.05309). A more comprehensive literature review is needed to clarify how this paper differs from those approaches.
- The authors claim that TopK SAEs produce more interpretable results than Ordered SAEs; however, they do not clearly quantify this difference (e.g., in terms of the percentage of explained latents).
- It is unclear whether the TopK and Ordered SAE results are directly comparable, given that one uses a dictionary size of 32× and the other 8× for computational reasons. Did the authors evaluate other dictionary sizes? How was the best size determined?
- It is somewhat unclear why the authors use two different methods for latent–concept identification. Why was the entire analysis not performed using the logistic regression coefficients, for example?
- The steering evaluation is somewhat limited, and the paper does not extend to steering properties such as developability or binding affinity.

Minor corrections
- Typo on page 7, Section 4.3: It should read “steering results in sequences that are novel, diverse, and antibody-like.”
- Typo in the meaningfulness statement: It should read “Life is organised through molecular sequences…”
- Figure 3: Clearly indicate which type of SAE architecture is represented in each plot. More generally, clearly specify the SAE type and model for each table and figure throughout the paper.
- In Section 4.1, why is the accuracy higher for the SAE latents than for the original hidden neuron activations? The paper should briefly comment on this and also report the hidden neuron accuracy for sequence-level features.
- Some sentences require clarification. For example, in the appendix:
  - “These results indicated that top latents encoded contextual information of the preceding residues.”
  - “Hence, activating one but not the others does not necessarily result in a predictable shift in model performance.” Do you mean model output?
  - “The functional importance of SAEs for studying LLMs lies in their ability to interrogate specific, domain-relevant concepts rather than an undefined set of all possible ones.” This claim should be clarified or supported with a citation.

---

### Meta-Review · Area_Chair_RkGN · 2026-02-27

**Recommendation:** Accept (Poster)
**Confidence:** 5

**Metareview:**

Accept.

---

### Decision · Program_Chairs · 2026-03-02

**Decision:**

Accept (Poster)

**Comment:**

Please see the meta-review.